# A Noninvasive Sweat Glucose Biosensor Based on Glucose Oxidase/Multiwalled Carbon Nanotubes/Ferrocene-Polyaniline Film/Cu Electrodes

**DOI:** 10.3390/mi13122142

**Published:** 2022-12-03

**Authors:** Yanfang Guan, Lei Liu, Shaobo Yu, Feng Lv, Mingshuo Guo, Qing Luo, Shukai Zhang, Zongcai Wang, Lan Wu, Yang Lin, Guangyu Liu

**Affiliations:** 1School of Electromechanical Engineering, Henan University of Technology, Zhengzhou 450001, China; 2Provincial Key Laboratory of Cereal Resource Transformation and Utilization, Henan University of Technology, Zhengzhou 450001, China; 3Department of Mechanical, Industrial & Systems Engineering, University of Rhode Island, Kingston, RI 02881, USA; 4School of Chemistry and Chemical Engineering, Henan University of Technology, Zhengzhou 450001, China

**Keywords:** sweat glucose, biosensor, enzyme-based electrode, noninvasive detection, electrochemical method

## Abstract

Diabetes remains a great threat to human beings’ health and its world prevalence is projected to reach 9.9% by 2045. At present, the detection methods used are often invasive, cumbersome and time-consuming, thus increasing the burden on patients. In this paper, we propose a novel noninvasive and low-cost biosensor capable of detecting glucose in human sweat using enzyme-based electrodes for point-of-care uses. Specifically, an electrochemical method is applied for detection and the electrodes are covered with multilayered films including ferrocene-polyaniline (F-P), multi-walled carbon nanotubes (MWCNTs) and glucose oxidase (GOx) on Cu substrates (GOx/MWCNTs/F-P/Cu). The coated layers enhance the immobilization of GOx, increase the conductivity of the anode and improve the electrochemical properties of the electrode. Compared with the Cu electrode and the F-P/Cu electrode, a maximum peak current is obtained when the MWCNTs/F-P/Cu electrode is applied. We also study its current response by cyclic voltammetry (CV) at different concentrations (0–2.0 mM) of glucose solution. The best current response is obtained at 0.25 V using chronoamperometry. The effective working lifetime of an electrode is up to 8 days. Finally, to demonstrate the capability of the electrode, a portable, miniaturized and integrated detection device based on the GOx/MWCNTs/F-P/Cu electrode is developed. The results exhibit a short response time of 5 s and a correlation coefficient R^2^ of 0.9847 between the response current of sweat with blood glucose concentration. The LOD is of 0.081 mM and the reproducibility achieved in terms of RSD is 3.55%. The sweat glucose sensor is noninvasive and point-of-care, which shows great development potential in the health examination and monitoring field.

## 1. Introduction

Diabetes is still one of the major problems endangering human health [1]. Moreover, the number of global diabetes patients has been increasing over years due to drastic changes of living habits in recent decades [2,3]. According to the estimation of the International Diabetes Federation (IDF), by 2045, the prevalence of diabetes among adults aged 18–99 will rise to 9.9% [4]. Such high incidence of diabetes certainly increases the social health burden and health care costs of households. Diabetic patients need continuous blood glucose monitoring [5], and possibly drug or insulin intervention in daily life.

Methods applied for glucose monitoring can be categorized into two classes: invasive and noninvasive. Generally, invasive detection methods use disposable test strips to collect blood samples squeezed from punctured skin or finger tips of the patients [6,7]. While such a method is simple and effective, repeating fingertip puncture can be a burden for the patients, not to mention the cost on the blood test strips and the medical wastes generated [8,9]. On the other hand, noninvasive methods allow painless blood glucose detection [10,11]. Electrochemical methods, electro-mechanical methods, optical methods and miscellaneous techniques have been explored and applied in this regard [11,12,13,14,15]. Malin used near-infrared light in a specific frequency range to irradiate the skin, as such blood collection is no longer required, and the absorbance spectrum is applied to measure the concentration of human blood glucose [16]. However, due to differences of infrared diffuse reflectance between individuals, errors exist in the calibration curve. Moreover, odor has also been explored for noninvasive glucose determination. For example, sweat of diabetes patients often emanates abnormal odor (e.g., fruity smell), thus it has been considered as a metric for diabetes diagnosis. In 2013, Olarte developed an electronic detection device for sweat glucose measurement [17]. The results show that the electronic odor system is sensitive to changes in glucose concentration even when the changes are in the order of a few mg/dL. However, the accuracy of the system at that resolution is low. Biosensors based on other types of platforms, for example, screen-printed electrodes [18] and bulk electrodes (CPE [19,20,21] or GCE [22,23]) are also being researched to detect glucose.

Indeed, compared with blood, body fluids such as sweat and tears are easier to obtain and do not impose painful sample collection to patients; thus, these methods hold great promise for future noninvasive diabetes monitoring. In particular, recent studies show that human sweat contains a variety of chemical substances, such as lactic acid, glucose, ethanol, ammonia, urea, etc. [5,12,14,24]. Among them, the sweat glucose concentration clearly reflects the concentration of blood glucose with a positive proportionality [25]. Therefore, a glucose sensor based on human sweat has opened up a new way for at-home, wearable and noninvasive glucose monitoring [26], as evidenced by many works reported in recent years. Vaquer et al. developed a colorimetric wearable biosensor that determines the glucose concentration in the test subject by comparing the color with a standard color chart [27]. Though the method is intuitive and simple, results are largely dependent on the discretion of the users and influence of ambient light; thus, this method is not suitable for precise quantification. Han designed a micro-electric sensor based on Pt-poly (L-lactic acid) (Pt-PLA) to measure the glucose in sweat with high accuracy [28]. Moreover, Katseli et al. created a 3D printed ring-shaped sweat glucose detector with a Pt-PLA electrode at its core [29]. The limit of glucose detection of this device was 1.2 μM in theory. 

Despite extensive progress made for sweat glucose sensing over the years, many challenges still exist. As the concentration of glucose in human sweat is from 0.06–0.2 mM and corresponds to 3.3–17.3 mM in blood glucose [12,30], glucose level observation in sweat is extremely difficult in view of its low concentration; therefore, needs highly sensitive devices are needed. Improvement of sensitivity remains a significant challenge when compared to that of the blood glucose sensors. To meet the demand, many methods have been used, such as changing the three-dimensional structure of the electrode [31], improving the immobilization efficiency of glucose oxidase [32,33,34], changing the charge transfer mechanism [35] and special processing of materials to enhance conductivity and so on. 

Glucose oxidase (GO_X_) is a well-characterized glycoprotein which catalyzes the oxidation of D-glucose to D-gluconolactone and hydrogen peroxide using molecular oxygen as an electron acceptor. It is widely applied in chemical, pharmaceutical, food, beverage, clinical chemistry, biotechnology, etc. [36,37,38]. The applications of GO_X_ in glucose detection biosensors have increased the demand in recent years [5,11,37,39]. Enzyme immobilization can improve the stability, reusability, and cost-effectiveness of catalytic enzymes [40]. Currently, a variety of materials have been used to immobilize glucose oxidase, such as graphene nanomaterials [41,42], polyethylene-g-acrylic acid [43], membrane bioreactor [44,45], carbon nanotubes [46], conductive polymers [47], etc. These materials usually have good mechanical strength [48,49], thermal stability and chemical stability [50], with large specific surface area and porous structure, thus are capable of loading more enzymes and intermediates [51,52,53,54,55]. However, limited charge-electron transfer of enzymatic reaction, high-cost of used noble metals and difficult preparation are still challenges [56,57,58]. To solve the abovementioned problems, the strategy that we have adopted is reasonably designing and manipulating a low-cost composite electrode with Cu substrates to modify the charge-electron transfer route, and further integrating biofunctional detection of glucose into one platform.

In this paper, we propose a novel portable, miniaturized and integrated sweat glucose detection device based on the GOx/MWCNTs/F-P/Cu electrode. The biosensor fulfills with the new three-dimensional anode, showing high sensitivity, low fabrication cost, more convenience and high enzyme immobilization. The detection device is highly integrated and does not need external auxiliary equipment. The glucose concentration can be obtained through an LCD screen, which is very convenient. Through experimental detection, the detection device designed by our group has high detection accuracy. In particular, it can obtain the glucose content in the human body through noninvasive detection, which shows the broad prospect of the application of this detection device in the field of health examination and monitoring in the future.

## 2. Experimental

### 2.1. Materials and Instruments

Materials: All reagents were applied without additional purification. Specifically, aniline (C_6_H_7_N) and chitosan (CS) were obtained from Macklin Biochemical Co., Ltd. (Shanghai, China). Hydrochloric acid (HCl, 36.5 wt%), nitric acid (HNO_3_) and isopropanol (C_3_H_8_O) were purchased from Yongfei Chemical Reagent Co., Ltd. (Shaowu, China). Ferrocene (C_10_H_10_Fe) and glucose anhydrous (C_6_H_12_O_6_) were obtained from ZhiYuan Reagent Co., Ltd. (Tianjin, China). Bovine serum albumin (BSA) phosphate buffer saline (PBS) and glutaraldehyde solution (CHO(CH_2_)_3_CHO, 5 wt%) from Phygene Biotechnology Co., Ltd. (Fuzhou, China). MWCNTs were purchased from Tanfeng Tech. Inc. (Suzhou, China). 5 wt% DuPont Nafion solution from Sigma-Aldrich lab & production materials Co., Ltd. (Shanghai, China). Glucose oxidase (GOx) was bought from Ekear Biotechnology Co., Ltd. (Shanghai, China).

Instruments: The MWCNTs were dispersed by an ultrasonic cleaner (Skymen Cleaning Equipment Shenzhen Co., Ltd., Shenzhen, China). The morphology and microstructure of F-P film, MWCNTs and anode were analyzed by a scanning electron microscope (SEM, Quanta 250 FEG, Hillsboro, AZ, USA). The voltage-current curve was measured by an electrochemical analyzer (CHI660E, Chen hua Instrument Co., Ltd., Shanghai, China). 

Process: The Cu substrate was prepared first. Then the Cu electrode was coated by ferrocene-polyaniline film (F-P film), MWCNTs and GOx successively. The next step is to fabricate the detection chip. The last step is to design and fabricate the paper-based enzyme biosensors (PEB) for glucose detection from human sweat. The detailed process is as follows.

### 2.2. Electroplating of Ferrocene-Polyaniline Film (F-P Film)

As shown in Figure 1a, the Cu substrate after cutting was first polished with sandpaper, then immersed in 15 mL HCl for 20 min. Next, the Cu substrate was washed by deionized water and blow-dried. Then, 0.028 g ferrocene and 1.08 g HNO_3_ were mixed together totally, after that a PBS buffer solution was added into the aforementioned mixed solution until the volume of the mixed solution was 12.5 mL. Afterward, 1.825 g HCl and 0.2325 g aniline solution were mixed together and stirred until the lumps were dissolved. Then, the PBS solution was also added dropwise into the mixed solution until the volume reached 12.5 mL as shown in Figure 1b. Several groups of comparative tests were conducted to explore the effect of separate electroplating and mixed electroplating in an electrolytic solution. The Cu substrate was immersed in the electrolyte solution. The parameters of the CV curve were set as 10 scanning cycles at a speed of 0.02 V/s between −0.3 V and +0.3 V. The polyaniline film was prepared when the film was brown. At the same time, ferrocene was modified on the electrode, which was used as the electron transfer intermediate. Then, the Cu electrodes coated by the F-P film were immersed in 2.5% glutaraldehyde water solution at 37 °C for 30 min. Finally, they were dried at room temperature.

### 2.3. Modification of Electrodes with MWCNTs and GOx

MWCNTs need to be dispersed before modifying the anode. The procedure is shown in Figure 1c. 3 mg MWCNTs, 0.25 mL nafion solution and 2.5 mL isopropanol solution were mixed together and dispersed in an ultrasonic cleaner for 3 h after short shaking. Next, 2 μL mixture was dropped on the surface of the Cu electrode coated with F-P film to modify the electrodes. Then, the anode was dried at room temperature for 6 h. After modification, the electrode was immersed in PBS and stored in a refrigerator at 4 °C. As shown in Figure 1d, 3 mg glucose oxidase and chitosan and glutaraldehyde solution were mixed together. Then the GOx mixture was dipped onto the electrode and put in the refrigerator for 4 h. The anode coated with GOx was fabricated for the detection of the glucose of the human sweat.

### 2.4. Fabrication Process of the Detection Chip 

The structure and parameters of the detection chip are shown in Figure 2. First of all, the hydrophilic and hydrophobic areas are printed on the filter paper using atom stamp printing technology. The detailed procedures are shown as follows. Figure 2a shows step 1 is to print the required figures on the acid paper. Step 2 is to expose the diagram on the stamp with a photosensitive seal machine as shown in Figure 2b. Step 3 is to put the stamp into the PDMS solution and immerse for 30 min, after exposing at 350 °C. Step 4 is to stamp the pattern on the filter paper for 2 min. Then, the filter paper is to be placed into a heater to heat until the hydrophobic channel is formed and the conductive silver glue is brushed onto it. The next step is to conceal the hydrophobic area by plastic film, which cuts the hydrophilic microchannel for the electrode and conductive layers by a laser engraving machine. Finally, the anode is packaged and the cathode on the reserved hydrophilic microchannel is glued using 3M glue, as shown in Figure 2g. 

### 2.5. Design and Fabrication of the Paper-Based Enzyme Biosensors (PEB) for Glucose Detection from Human Sweat

A portable, miniaturized and integrated detection device for real-time detection of human sweat glucose concentration is designed and developed. As shown in Figure 3, the detection device is embedded in a 3D printing shell, and the main parts include: power supply (3.7 V), A/D conversion module, signal amplifier, DC-DC module and LCD display module. In order to reduce the cost and power consumption, an STM32 A/D converter is used as the controller, and as many domestic components as possible are used, with a total weight of 200 g.

The detection process is as follows. Firstly, the encapsulated chip is inserted into the interface of the detection device. Next, an appropriate amount of sweat is dropped to the designated position of the chip. After waiting for 5 s, the power is turned on. A small amount of glucose in the sweat reacts with glucose oxidase on the electrode to produce a weak current *I*. The current passes through the conversion resistance *R*, and a conversion voltage *V_i_* is equal R times I. The *V_i_* passes through the in-phase amplification circuit, and an amplification voltage *V*_0_ is obtained. The *V*_0_ is also in the range of voltage required for A/D conversion (0–3.3 V). The obtained voltage *V*_0_ will be converted into a binary number, and a current value will be obtained. Then, the glucose concentration in the sweat will be obtained and displayed by an LCD display.

## 3. Results and Discussion

### 3.1. Characterization of F-P Film and MWCNTs

As shown in Figure 4a, b, it can be clearly found that the peak current of the CV curve of F-P film prepared by separate electroplating is about 30 mA, which is significantly higher than that of F-P film (about 10 mA) prepared by mixed electroplating. Figure 4c shows the CV curves of F-P film prepared by electropolymerization. It can be seen that obvious redox peaks appear in the process of electropolymerization. With the increase of scanning times, the peak potentials of redox peaks move from −0.19 V to +0.19 V, and the peak currents gradually change from ±50 mA to ±100 mA. By analyzing the CV curve, it can be seen that the voltage difference between the two peaks is 0.38 V, which shows that the electrolytic solution and CV scanning parameters used by our experimental team can well attach a layer of F-P film to the Cu substrate. Figure 4d is a 40-times electron micrograph of F-P film. It can be seen that the F-P film is brownish yellow and covered with golden particles. The modified MWCNTs were characterized by SEM. As shown in Figure 4e,f, after nafion dispersion and ultrasonic treatment, the carbon nanotubes are flocculent on the electrode, which increases the surface area of the working electrode and is conducive to improving the electron transfer rate and the immobilization effect of the enzyme.

### 3.2. Characterization of the Paper-Based Enzyme Anode

The electrochemical characteristics of the fabricated electrodes were tested before they were used in the sensor. Figure 5a shows the CV curves of different modified Cu electrodes (bare Cu, F-P/Cu, MWCNTs/F-P/Cu) at the same scanning rate and scanning range (−0.3 V to +0.3 V, 0.01 V/s). It can be seen that all electrodes have clear redox peaks in the range of −0.3 V to +0.3 V. With the increase of surface modifications, the redox peaks also increase, which indicates that the prepared F-P film and MWCNTs have good nanostructure and electrochemical properties, effectively improving the electron transfer rate. Figure 5b shows the CV curves of the final electrode (GOx/MWCNTs/F-P/Cu) in different concentrations of glucose solution (0, 0.1, 0.3, 0.7, 1.0, 1.3, 1.7, 2.0 mM). As shown in the figure, the size of the CV curve is obviously proportional to the glucose concentration, and the shape of all curves are almost the same, with clear redox peaks. With the increase of glucose concentration, these peaks showed a regular gradient slope, and there was no significant potential shift between the peaks. The peak currents of redox reaction gradually change from +4 mA and −3 mA to +10 mA and −11 mA. In order to optimize the working potential of the glucose sensor, six different potentials (0.10, 0.15, 0.18, 0.22, 0.25 and 0.30 V) were selected in the potential range of +0.1 V to +0.3 V, and the current response was tested in the glucose solution with a concentration of 1.0 mM by chronoamperometry. As shown in Figure 5c, with the increase of potential, the current responses increase gradually, and the strongest current signal response is observed at 0.3 V. However, it can also be seen that the current response at 0.3 V is not as smooth as other groups of curves, which may be due to the instability of the electrode performance at high potential. Therefore, we decided to use 0.25 V as the working potential of the sensor. In order to further study the current response at this potential, the chronoamperometry test was carried out on the electrode at 0.25 V, and the results are shown in Figure 5d. When 0.1 mM glucose was added into the PBS buffer at 100, 300, 500 and 700 s, the current response changed significantly within 10 s, which indicated that the electrode was highly sensitive to glucose concentration with good test results at 0.25 V. In order to evaluate the effective working time of the prepared electrode, the electrode was stored in the indoor environment, and its maximum current response in 1 mM glucose solution was measured and recorded daily for 20 days. As shown in Figure 5e, the maximum current in the first 8 days was relatively stable, and it began to decrease significantly until the 9th day, reaching 60% of the original maximum current on the 12th day and 45% after 20 days. It can be seen that the electrode has a long working time, and the effective working lifetime is up to 8 days.

### 3.3. Performance Analysis of the Biosensor in Human Sweat

In order to verify the feasibility, the detection device with the GOx/MWCNTs/F-P/Cu electrode was first used to measure the artificial glucose solution. We performed a quantitative glucose analysis using amperometric response. A micro current meter was used to measure the glucose solution with different concentrations ranging from 0 to 14 mM. Each concentration is measured at least ten times, the response current is recorded and the average of each group is calculated as the final desired result. As shown in Figure 6e, the current gradually increased with the increase in glucose concentration. The fitted curve shows the linear relationship between glucose concentration and response current with the correlation coefficient R^2^ of 0.9988. The linear regression parameters between glucose concentration and response current were used to determine the limit of detection (LOD). Using the equation LOD = 3 * σ/S, where σ is the estimated standard deviation of the minimum concentration detected and S is the slope of calibration curves, the LOD obtained was 0.081 mM. 

The next step is to write the sub-function into the system as the basis for detecting glucose concentration. In order to check the accuracy and repeatability of the detection device, the device is used to measure glucose solutions of different concentrations five times, respectively. The detection device exhibits a short response time (5 s) and good repeatability and reliability. Table 1 shows the comparison of the analytical performance of some reported glucose biosensors with GOx/MWCNTs/F-P/Cu biosensors. It can be seen that response time, linear range and LOD of GOx/MWCNTs/F-P/Cu are the same and even better than other reported glucose biosensors.

Then, we use the device to analyze glucose concentration in sweat. As shown in Figure 6a–d, thirteen volunteers’ (average age of 23 years old) sweat was sampled through exercise and glucose concentration was analyzed by the device. The blood glucose concentrations were analyzed simultaneously for comparison. The micro current meter was also used to measure glucose in sweat. The results were shown as Figure 6f. It can be observed that blood glucose concentration and response current of sweat vary greatly from volunteer to volunteer, while the trend is the same. Similarly, the linear regression equation in Figure 6g was fitted between blood glucose concentration and response current of sweat with the correlation coefficient R^2^ of 0.9847. Due to the difference in the corresponding relationship between the glucose concentration in sweat and the blood of different individuals, the linear correlation coefficient is not high enough. The reproducibility achieved in terms of RSD associated with the slope of the calibration curve was 3.55%. Then, the linear regression equation is written into the detection system. Through measuring sweat, we can obtain blood glucose concentration data as shown in Figure 6d. The measurement result indicates that the glucose concentration obtained from sweat by our detection device is consistent with blood glucose concentration. The portable, miniaturized and integrated detection device can meet the requirements of glucose concentration detection in sweat. 

## 4. Conclusions

In this paper, we proposed an enzyme-based electrode composed of GOx/MWCNTs/F-P/Cu, which was used as the anode in a biosensor to detect the glucose in human sweat. The method is a noninvasive and point-of-care glucose detection. The electrode was tested by CV and chronoamperometry. The result shows the electrode has a wide glucose detection range of 0-14 mM with an optimal current response at 0.25 V, which suggests that the electrode could meet the requirements of sweat glucose detection. A portable, miniaturized and integrated detection device was fabricated to analyze glucose in human sweat, exhibiting a short response time (5 s) and high reproducibility (RSD = 3.55%). The main advantages of the sensor include: (1) Cu was used as the electrode substrate, which greatly reduced the cost of fabrication; (2) the unique three-dimensional structure GOx/MWCNTs/F-P/Cu was used, which greatly improved the efficiency of enzyme immobilization; (3) the device is portable, highly sensitive and accurate in detecting glucose in human sweat. The sweat glucose sensor presented in this paper shows great development potential in the health examination and monitoring field.

## Figures and Tables

**Figure 1 micromachines-13-02142-f001:**
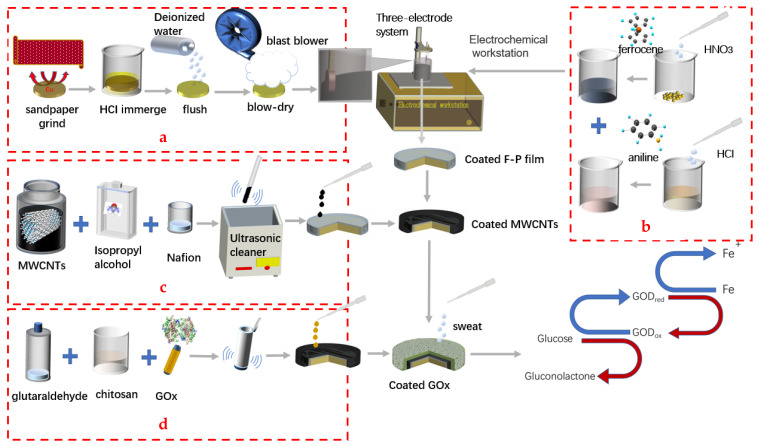
Characterization and fabrication process of paper-based enzyme biosensors coated with F-P film, MWCNTs and GOx. (**a**) step 1: treatment of Cu sheet. (**b**) step 2: preparation of electrolytic solution. (**c**) step 3: dispersion of MWCNTs. (**d**) step 4: preparation of enzyme solution.

**Figure 2 micromachines-13-02142-f002:**
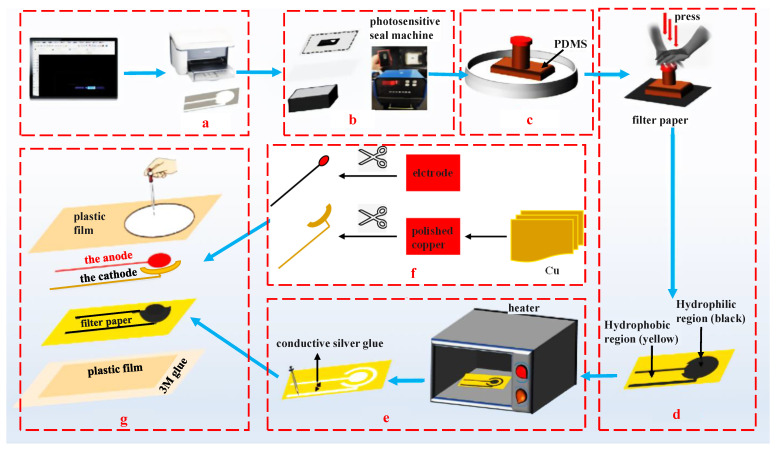
The fabrication process of the electrodes. (**a**) step 1: printing hydrophilic channels on the paper. (**b**) step 2: exposing the diagram. (**c**) step 3: immersing the stamp into PDMS. (**d**) step 4: stamping the pattern on the filter paper. (**e**) step 5: brushing the conductive silver glue. (**f**) step 6: fixing the cathode and anode. (**g**) step 7: packaging the whole detection chip.

**Figure 3 micromachines-13-02142-f003:**
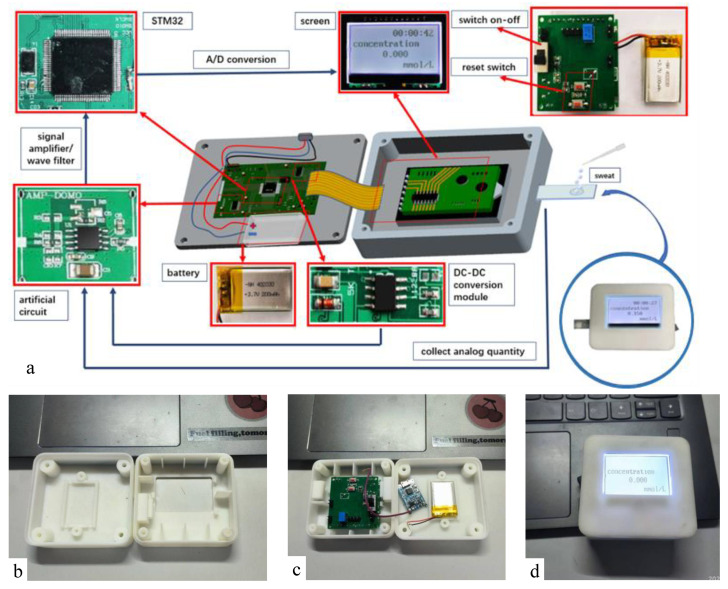
The portable sweat detection device: (**a**) internal structure diagram. (**b**) 3D printing shell. (**c**) real internal photo. (**d**) real detection device photo.

**Figure 4 micromachines-13-02142-f004:**
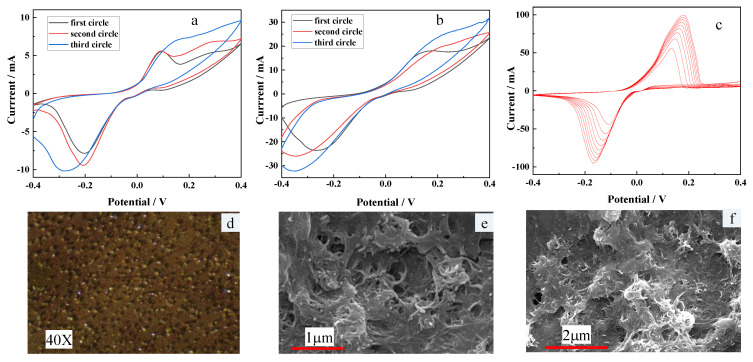
The results of the experiment. (**a**) CV curves of F-P film prepared by mixed electroplating. (**b**) CV curves of F-P film prepared by separate electroplating. (**c**) CV curves of F-P film prepared by electropolymerization. (**d**) micrograph of F-P film after electroplating. (**e**,**f**) SEM photographs of MWCNTs after dispersion.

**Figure 5 micromachines-13-02142-f005:**
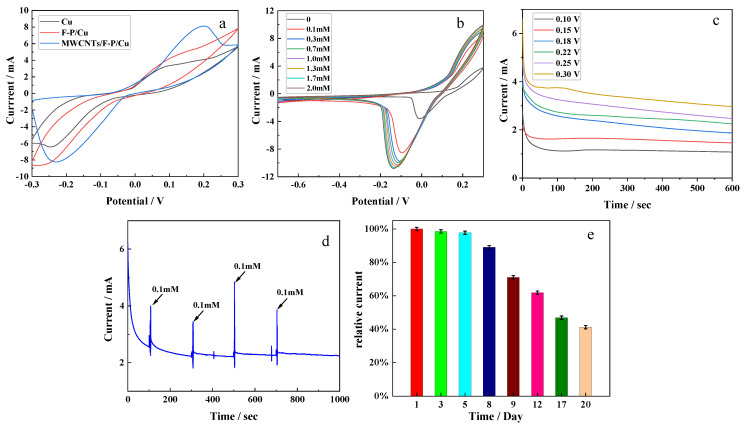
Electrochemical characterization of the fabricated anode. (**a**) CV curves of Cu electrodes with different modifications (bared Cu, F-P/Cu, MWCNTs/F-P/Cu). (**b**) CV curves of the fabricated electrode (GOx/MWCNTs/F-P/Cu) in glucose solution of different concentrations. (**c**) Chronoamperometry curves of the fabricated electrodes (GOx/MWCNTs/F-P/Cu) at different potentials. (**d**) Chronoamperometry curve of the fabricated electrode at best potential (0.25 V); (**e**) Maximum currents of the fabricated electrodes after several days in indoor environment.

**Figure 6 micromachines-13-02142-f006:**
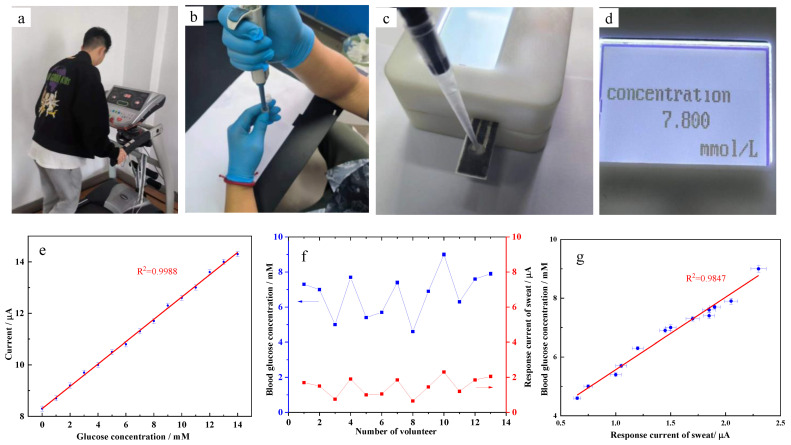
The experiments of sweat sampling and glucose detection. (**a**) exercise for sweat sampling. (**b**) sampling with a pipette. (**c**) analysis with the device. (**d**) the analysis result displaying on LCD. (**e**) The relationship between response current and glucose concentration of artificial glucose solution. (**f**) The results of volunteers’ blood glucose concentration and response current of sweat. (**g**) The correlation degree between response current of sweat and blood glucose concentration.

**Table 1 micromachines-13-02142-t001:** Comparison of the analytical performance of some reported glucose biosensors.

Electrode Platform	Response Time (s)	Linear Range (mM)	LOD (μM)	Ref.
GOD/MAA/AuNPs/TiO_2_NT/Ti	<10	0.40–8	310	[59]
GOx-PoPD/AuNPs-GO/Pt	10	0.1–3.8	75	[60]
GOx/P-L-Arg/f-MWCNTs/GCE	<5	0.004–6	0.1	[61]
Cs/GOx/PGA/GCE	-	0.5–5.5	120	[62]
GOD/AuNPs-MoS_2_/Au	-	0.25–13.2	0.042	[63]
GOD/GA/MLN/GCE	-	1.0–135	100	[23]
GOx/PAA/CPE	4	0.05–1	69.2	[19]
GOx/MWCNTs/F-P/Cu	<5	0–14	81	This work

GOD, GOx: glucose oxidase; MAA: mercaptoacetic acid; AuNPs: Gold nanoparticles; TiO_2_NT: TiO_2_ nanotube array electrode; Ti: Titanium foil; PoPD: poly-o-phenylenediamine; GO: graphene oxide; P-L-Arg: poly(L-arginine); f-MWCNTs: functionalized multiwalled carbon nanotubes; GCE: glassy carbon electrode; Cs: chitosan; PGA: poly(glutamic acid); GA: glutaraldehyde; MLN: molybdenite; PAA: poly(L-glutamic acid); CPE: carbon paste electrode; F-P: ferrocene-polyaniline.

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
