# Peer review of "A Noninvasive Sweat Glucose Biosensor Based on Glucose Oxidase/Multiwalled Carbon Nanotubes/Ferrocene-Polyaniline Film/Cu Electrodes"

_micromachines, 2022, doi:10.3390/mi13122142_

Round 1

Reviewer 1 Report

Title: A noninvasive sweat glucose biosensor based on glucose oxidase / multiwalled carbon nanotubes / ferrocene-polyaniline film / Cu electrodes

Major revision

1. Authors used multiwalled carbon nanotubes / ferrocene-polyaniline film / Cu electrodes for sensing of glucose. Authors must explain selectivity towards uric acid in sweat. 

2. Authors must add few more charactersation aspects of composites.

3. Electrochemical impedance spectroscopy is more important in enzymatic sensors. authors must add this analysis.

4. Authors should polish the language of the manuscript. The English of the manuscript needs deeply revision. There are some grammatical and typo errors in the text that need to be re-checked and corrected more carefully.

5. Figures and captions should be check and verify overall manuscript content.

6. Discussion/Conclusion: I was surprised to see that the method, according to the manuscript (discussion & conclusion), has no major limitations. Is this correct? If not, please specify at the end of the Discussion.

Reviewer 2 Report

Micromachines-1882592

This work presented a cost-effective noninvasive sweat glucose biosensor based on glucose oxidase multiwalled carbon nanotubes-ferrocene-modified electrodes. The topic is very important however, it is not the hottest topic nowadays in the biosensors field. I find some important issues.
